Original research

# Matching study using health and police datasets for characterising interpersonal violence in the community of Khayelitsha, South Africa 2013–2015

Ardil Jabar ,[1,2] Tolu Oni,[1,3] Leslie London ,[1] Annibale Cois ,[4] Richard Matzopoulos[1,2]

[1]School of Public Health and Family Medicine, University of Cape Town, Cape Town, South Africa
[2]Burden of Disease Research Unit, South African Medical Research Council, Cape Town, South Africa
[3]MRC Epidemiology Unit, University of Cambridge, Cambridge, UK
[4]Department of Global Health, Division of Health Systems and Public Health, Faculty of Medicine and Health Sciences, Stellenbosch University, Stellenbosch, South Africa

**Correspondence to**
Dr Ardil Jabar;
Ardil.Jabar@uct.ac.za

## ABSTRACT

**Objectives** The Cardiff Model of data sharing for violence prevention is premised on the idea that the majority of injury cases presenting at health facilities as a result of interpersonal violence will not be reported to the police. The aim of this study was to determine the concordance between violent crimes reported to the police with violence-related injuries presenting at health facilities in Cape Town, South Africa.

**Methods** We conducted a retrospective analysis of secondary cross-sectional health and police data, from three health facilities and three police stations in the community of Khayelitsha, Cape Town. 781 cases of injuries arising from interpersonal violence seen at health facilities were compared with 739 violence-related crimes reported at police stations over five separate week-long sampling periods from 2013 to 2015. Personal identifiers, name and surname, were used to match cases.

**Results** Of the 708 cases presenting at health facilities, 104 (14.7%) were matched with police records. The addition of non-reported cases of violence-related injuries from the health dataset to the police-reported crime statistics resulted in an 81.7% increase in potential total violent crimes over the reporting period. Compared with incidents reported to the police, those not reported were more likely to involve male patients (difference: +47.0%; p<0.001) and sharp object injuries (difference: +24.7%; p<0.001). Push/kick/punch injuries were more frequent among reporting than non-reporting patients (difference: +17.5%; p<0.001).

**Conclusion** These findings suggest that the majority of injuries arising from interpersonal violence presenting at health facilities in Khayelitsha are not reported to the police. A data-sharing model between health services and the police should be implemented to inform violence surveillance and reduction.

## INTRODUCTION

The Cardiff Model employs the strategic use of information from the health sector to improve policing and is an example of cross-sectoral collaboration. Developed by the Violence Research Group at Cardiff University, the model provides a way for communities to gain a clearer picture about where

### STRENGTHS AND LIMITATIONS OF THIS STUDY

⇒ To the best of our knowledge, this is the first study to attempt direct name matching between the health and the police to quantify the level of under-reporting of interpersonal violence-related incidents.
⇒ Cross-sectional violence-related data were collected from three health facilities and three police precincts within the community of Khayelitsha over a 3-year period.
⇒ One of the limitations of this study is that patients who presented to a health facility but only lodged criminal cases later outside of the 1-week study periods over the 3 years would not be included in the police dataset and therefore not be included in the matching process.

violence occurs by mapping both police and hospital violence data in combination.[1] Application of the model in Cardiff in 2011 found that one-half to two-thirds of violence which results in hospital treatment is not known to the police.[2] Subsequent research found that police recording of violence was limited to people reporting violent offences, but that many of the injured who presented at emergency departments (EDs) for treatment do not report the incident to police because they are either afraid of reprisals, do not want their own conduct scrutinised, or they do not think the reporting will result in effective police action.[3]

These findings have been replicated in other settings including England and the USA, and the model is being considered for implementation in Australia and Jamaica.[4,5] Similarly, findings from the US Department of Justice National Crime Victimization Survey, a national household survey that sampled 701 000 individuals 12 years and older from 2006 to 2010, revealed that more than half (52%) of all violence incidents in the USA were not reported to law enforcement.[6] We

were unable to identify any similar studies in South Africa that compared the reporting of violence-related injuries to the police versus health facilities. However, we do know that violence is not always reported to the police. For example, in a community survey in Khayelitsha only 49% of violent incidents between family members were reported to the police.[7]

The Western Cape Safety Plan, a policy document developed by the Western Cape Government, advocates the use of data and technology to understand violent crime patterns to inform the deployment of law enforcement resources and investigators accordingly and furthermore acknowledges research and analysis as an important component of its evidence-based policing (EBP) strategy.[8] The policy document lends support to the implementation of EBP interventions such as the Cardiff Model locally.

The aim of this study was to determine the concordance between violent crimes reported to police stations and cases of injuries arising from interpersonal violence presenting to health facilities. Secondary objectives included the estimation of an adjusted crime profile (where police-reported crime statistics included unreported violence from the Health data) and a comparison between patient populations that reported violence to the police with those that did not.

## METHODS
### Study setting
Khayelitsha is a periurban low-income community of approximately 400 000 people located 30 km from the Cape Town city centre.[9] The homicide rate is well above the national average of 31 murders per 100 000 population at between 76 and 108 per 100 000 at Khayelitsha's different police stations.[10] The community has mixed housing (informal and approximately 45% formal dwellings) and high unemployment (36%)[9] and is serviced by three police stations: Khayelitsha, Harare and Lingelethu West. Trauma cases are treated at three public-sector health facilities: Khayelitsha District Hospital (KDH), Site B and Michael Mapongwana Community Health Centre (figure 1).

### Study design
This was a retrospective cross-sectional analysis of secondary health and crime data from Khayelitsha for five 1-week-long periods collected between September 2013 and October 2015.

We assessed whether people presenting with injuries arising from interpersonal violence at health facilities went on to report these incidents to the police and whether victims reporting a crime to the police sought treatment at a health facility. We quantified this using personal identifier matching.

### Data sources
The health data used in this study were originally collected by a non-profit research organisation, the Health Systems

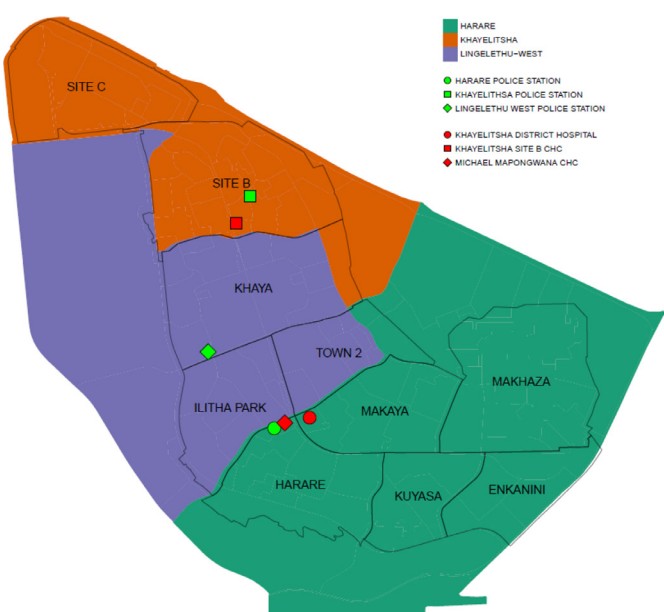

**Figure 1** Participating health facilities and police stations in relation to study areas in the community of Khayelitsha.

Trust (HST), for a study commissioned by the Western Cape Department of Health (DOH). Data were collected for five 1-week-long cross-section periods between 2013 and 2015. The health data were collected via face-to-face interviews with patients with violence-related injuries using a standardised data collection tool loaded on an electronic tablet, in the three health facilities (figure 1) from 2013 to 2015.[11]

All emergency cases presenting to the health facilities were assessed for acute injuries which were classified according to cause of injury, namely violence, transport, other unintentional injury or self-harm. Retrospective folder reviews of sexual assault cases that presented to the clinical forensic units at KDH during the corresponding study period were also included.

The police dataset included all violent crimes reported at all three police stations over the same time periods as the health dataset. These data were recorded in the national South African Police Services (SAPS) Crime Administration System in Pretoria. A formal request to access the data were made to the SAPS National Research Division, which was granted (Reference no 3/34/2) on 24 November 2017.

### Patient and public involvement
No patients were involved in the study.

## VARIABLES
Cases presenting to health facilities for violence-related injuries included the following variables: hospital folder number as a personal identifier; age; gender; date and time of injury event; instrument used, type of violence; area (incident location); triage score and alcohol use.

The inclusion of patient hospital numbers allowed for accessing patients' details on the Clinicom hospital

system (a hospital patient tracking information system)[12] to identify patient names for matching with police data.

All acute injuries classified as person-on-person intentional violence were described according to the *external cause of injury* as per the International Classification of Diseases (ICD 10), tenth revision,[13] that is, sharp object (ICD10 code X99), blunt object (Y00), firearm, push/kick/punch (Y04), human bite (Y08), explosion (X09), choking/strangulation (X91), fire burn (X97), other burn, poisoning (X85) or unknown; and *type of violence* defined in the study as rape/sexual, gang-related, property crime-related, interpersonal or unknown. 'Gang related' and 'unknown' were collapsed into a combined 'interpersonal violence' category for purposes of comparison with the police data, which did not include these categories.

Cases recorded in the 'unknown' category included cases that recorded a specific cause of injury, where the type of violence was marked unknown and the victim-perpetrator category was also marked unknown. Patients treated for burns of unknown cause were included in the 'other burn' category.

The location of each assault was self-reported by patients attending health facilities according to 10 predefined study areas (see figure 1). The collection of incident location data allowed for geolocating each case in 1 of the 10 study areas: Enkanini, Harare, Ilitha Park, Khaya, Kuyasa, Makaya, Makhaza, Site B, Site C and Town 2 (figure 1). Cases that were geolocated beyond the 10 areas were excluded from the analysis.

The health data also included additional variables of interest that were not available in the police data. The triage colour provided a proxy for injury severity, namely red (refer to major area for emergency management (immediate)), orange (refer to major area for urgent management (target time <10 min)), yellow (refer to major area for urgent management (target time <1 hour)), green (refer to designated area for non-urgent management (target time <4 hours)) and blue (refer to doctor for certification (target time <2 hours)). Alcohol and other drug use for each case were assessed by clinical judgement. Volume of consumption and consumption beyond 6 hours of the injury event was not assessed.

The police data included name and surname as personal identifiers, age, gender, date and time of crime event, type of crime, instrument use and incident location. Types of crimes included assault with the purpose to inflict grievous bodily harm, attempted murder, common assault, domestic violence, rape, sexual assault and all types of robbery, which involve an interpersonal interaction with an implicit threat (ie, common robbery, house robbery, robbery with a weapon or instrument other than a firearm, robbery and attempted robbery with a firearm, robbery and attempted robbery at business premises, carjacking (stealing an occupied motor vehicle)).[14]

For purposes of comparison with the *type of violence* recorded in the health data, crime type was coded as follows: rape and sexual assault cases (rape/sexual);

common assault, assault with the purpose to inflict grievous bodily harm, domestic violence attempted murder (interpersonal violence); all types of robbery, which involve an interpersonal interaction with an implicit threat (ie, common robbery, house robbery, robbery with a weapon or instrument other than a firearm, robbery and attempted robbery with a firearm, robbery and attempted robbery at business premises, carjacking (stealing an occupied motor vehicle) (property crime related)).

The police recorded instrument use as a separate field and the categories were broadly consistent with the health data, as follows: sharp instrument, glass, key, knife, panga (bladed weapon), bottle, bottle head, screwdriver (sharp object); blunt instrument, hammer, iron pipe, brick, stick, stone/brick (blunt object), firearm, pistol, revolver (firearm), head, fist, body part, feet, hands (push/kick/punch), belt, open hands (choking/strangulation); tyre, matches, petrol bomb (fire burn); electricity, ammunition (other burn); tik (methamphetamine), mandrax pills (quaalude), dagga (cannabis) (poisoning) and cell phone, motor vehicle, vehicle, cash, money, bank card, unknown (unknown).

Police incident location data were mapped based on self-reported data provided by victims of violence and were either geolocated using XY coordinates or recorded as a street address.

### Case selection

Police records with multiple victims were expanded by replicating the common information (place, type of crime, etc.) to generate a single record for each victim in the final dataset.

Cases that were excluded from the health data included medical cases not related to injuries and injuries not caused by violence perpetrated on the victim by another subject (Condition 1); events that occurred outside the 10 areas of Khayelitsha (Condition 2); cases in which the victim was younger than 16 years (Condition 3) (age of sexual consent in South Africa) and events that occurred outside the time period specified (Condition 4).

Cases that were removed from the police data include cases other than the types of crimes described above (Condition 1); events that occurred outside the 10 areas of Khayelitsha (Condition 2); cases in which the victims were younger than 16 years (Condition 3) and events that occurred outside the time period specified (Condition 4).

### Data analysis

The two datasets were captured in Microsoft Excel and imported into the R Statistical environment V.3.5.2 for analysis.[15] Two-side statistical tests with a conventional 5% cut-off for statistical significance were conducted to assess differences in characteristics between injury patients from the health database and crime victims from the police database, and between reporting and non-reporting patients in the health database.

$\chi^2$ tests were used to evaluate differences in the distribution of categorical variables. When comparing the police

dataset with the health dataset, a robust estimator of the test variance was used to take into account the partial overlapping of the two samples.[16]

Cases from health facilities were matched with police data using name, surname, gender and age as matching keys. Some latitude was accorded to allow for spelling variations and approximate recording of ages. This was done as follows: patient hospital numbers were entered into the Groote Schuur Hospital Clinicom database to retrieve the patient's name. The Levenshtein distance (a string metric for measuring differences between two sequences of characters[17]) was calculated for each patient between his/her name and each of the names present in the police dataset, and the police record(s) was selected corresponding to the minimum distance as 'candidate' matches. In the second step (exclusion of unlikely matches), the potential matches generated from the first step were retained for subsequent analysis if (1) names were identical between the two datasets; (2) the Levenshtein distance was 1, gender was the same and reported ages differed less than 5 years or (3) the Levenshtein distance was >1, gender was the same and reported ages differed less than 3 years. In the third step (manual selection), the potential matches resulting from the previous step were manually inspected to verify the actual correspondence between victim names and the consistency of the reported time of the violent episode, which led to the elimination of further cases.

An estimation of an adjusted crime profile was done by adding unreported patient violence seen at health facilities to the police-reported crime statistics. The patients who reported violence to police (reporting patients) and the patients who did not report the violence to the police (non-reporting patients) were inferred from the matching process and are described with regards to demographic characteristics (age and gender), instrument use, type of offence, triage code and alcohol use.

### Ethics

Ethical permission to conduct this research was granted by the University of Cape Town Human Research Ethics Committee (UCT HREC 861/2016) and the South African Police Services Research Division (3/34/2). With regard to the HST data, informed consent was obtained only from violence-related injury cases and where given, these individuals were interviewed by the data collectors, with only violence-related injury cases having the full data collection tool completed. Informed consent was granted by Groote Schuur Hospital for permission to access the Clinicom patient database to complete the personal identifier matching. The study Biostatistician received both health and police datasets, conducted the identifier matching process and deleted all data permanently following this process. By aggregating the data, removing personal identifiers, geolocating cases to residential suburb and not street address and employing a 1 month delay of data dissemination, every technical effort is made to prevent the identification of individual cases and protect individual identity.

## RESULTS

### Summary of included data

The health dataset initially included a total number of 9608 records of patients who presented at the three facilities during the five time periods considered. After applying exclusions (medical conditions, injury not due to interpersonal violence, injury occurring out the study area or study period and child injury), the final dataset included 781 records (figure 2).

The police dataset initially included a total number of 2161 records of criminal episodes. After applying exclusions (cases occurring outside the five study periods, cases not included in the types of included crimes list, cases where the victim was younger than 16, cases where the incident location was outside 1 of the 10 study areas), the final dataset included 739 records (figure 2). Three hundred and fifty-eight police records had missing values on gender and age.

Patients seen at health facilities (69.8%) and victims of violence (51.4%) were both more frequently males (table 1). The mean age for patients was 29.9 for patients and 35.2 for victims of violence. The most common age ranges for both datasets were 25–34 (health 42.2%, police 34.1%), 16–24 (health 33.7%, police 22.3%) and 35–44 (health 15.1%, police 18.9%). At health facilities, violence peaked on Sundays (33.7%), whereas violent crimes were most frequently reported to the police on Saturdays (22.3%).

The most common instrument described in the police database was an assault with body part (push/kick/punch) followed by unknown, firearm, sharp object and blunt object. At health facilities, the most common instrument use was sharp objects, followed by an assault with body part (push/kick/punch), blunt objects, unknown and firearms. Sharp objects comprised 60.7% for health presentations versus 12% for police presentations, while body part was 41.8% for police station presentation, while only 14.7% for health.

Site B recorded the most cases in both health (36.2%) and police (32.9%) datasets with Ilitha Park (2.7%) and Enkanini (1.2%) recording the fewest cases, respectively.

### Concordance

Among the 708 patients being treated for violence-related injuries at health facilities for which the name and surname could be retrieved from the Clinicom database, only 104 reported the incident to the police, which equates to a matching ratio of 14.7% (figure 3).

### Adjusted crime profile

Combining unreported cases from health records to the police-reported crime statistics provides an adjusted crime profile (table 2). The addition of patient violence from the health data to the police-reported crime

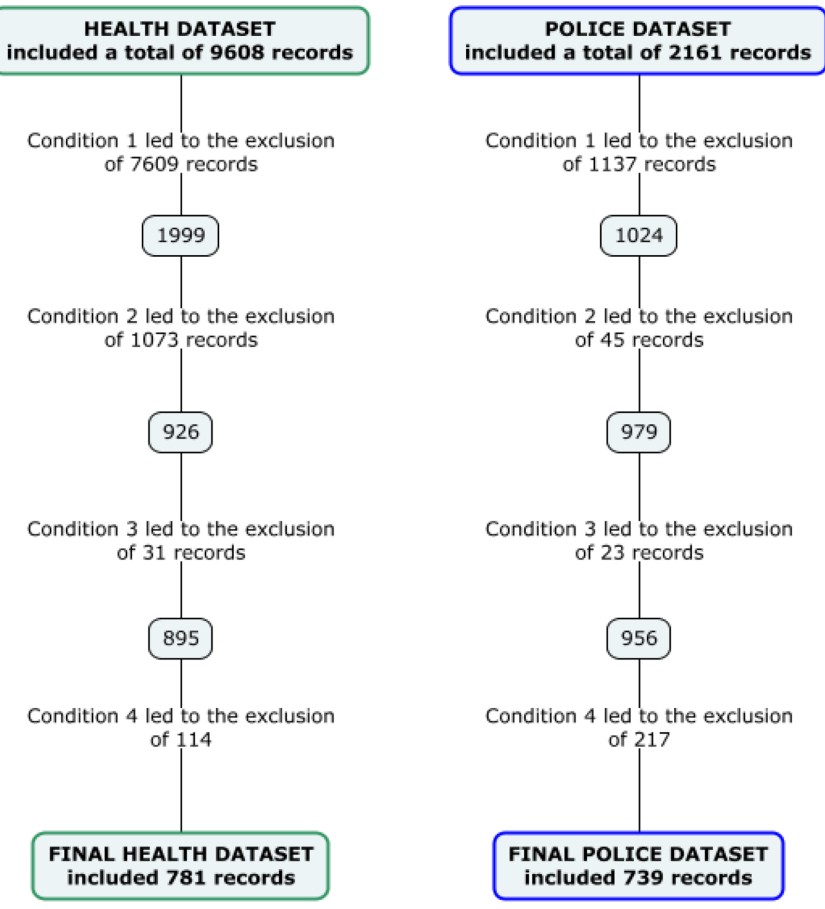

**KEY:**

Health condition 1: Cases that were excluded from the health data included medical cases not related to injuries and injuries not caused by violence perpetrated on the victim by another subject

Police condition 1: Cases that were removed from the police data include cases other than the types of crimes described in variables section

Condition 2: events that occurred outside the ten areas of Khayelitsha

Condition 3: cases in which the victim was younger than 16 years

Condition 4: events that occurred outside the time period

**Figure 2** Case selection flow chart for health and police datasets.

statistics resulted in an 81.7% increase in potential total violent crimes over the reporting period. Males represent the majority in the combined (65.0%) dataset. In terms of day of week, Saturday and Sunday were the busiest days in the combined dataset with Thursday being the quietest.

Push/kick/punch (41.8% and 40.3%) is the most common instrument used in the police and combined dataset, respectively, with the the most common type of violence being interpersonal violence for both (57.1% and 58.1%). Site B, Site C and Makhaza ranked among the top three for both datasets with Enkanini (1.5%) similarly recording the fewest cases.

Notable shifts in ranking when combining the datasets were found in day of week (Saturday ranked 1 in police vs 2 in combined dataset), time of day, instrument used (push/kick/punch ranked 1 in police data (41.8%) and sharp object ranked 1 in combined data (35.4%)), and area where the positions for Makhaza and Site C were changed in the top three ranked areas.

### Predictors of non-reporting

Male patients (73.5%) who were stabbed (64.1%), had alcohol use confirmed or suspected (58.9%) and experienced crime (39.6%) or gang-related incidents (11.4%) were less likely to report their injuries to the police. Patients suffering sharp object (39.4%) or push/kick/punch injuries (29.8%) or sexual assault/rape (8.7%) were more likely to report their injuries (table 3).

No statistically significant differences were present regarding age and severity of the injury (as measured by

**Table 1** Distribution of cases recorded in health facilities and by the police according to gender, age, day of the week, time of day, instrument use, type of violence, incident location, triage score and alcohol/drug use

| Variable | Health data (N=781) n (%) | Police data (N=739) n (%) |
|---|---|---|
| Name and surname completeness | 708 (90.7) | 739 (100) |
| Gender* | | |
| Males | 545 (69.8) | 196 (51.4) |
| Females | 236 (30.2) | 185 (48.6) |
| Age (years)* | | |
| 16–24 | 263 (33.7) | 85 (22.3) |
| 25–34 | 330 (42.2) | 130 (34.1) |
| 35–44 | 118 (15.1) | 72 (18.9) |
| 45–54 | 46 (5.9) | 56 (14.7) |
| 55+ | 24 (3.1) | 38 (10.0) |
| Day of week | | |
| Monday | 95 (12.2) | 103 (13.9) |
| Tuesday | 50 (6.4) | 72 (9.7) |
| Wednesday | 48 (6.1) | 75 (10.1) |
| Thursday | 45 (5.8) | 64 (8.7) |
| Friday | 77 (9.8) | 100 (13.5) |
| Saturday | 203 (26.0) | 165 (22.3) |
| Sunday | 263 (33.7) | 160 (21.7) |
| Time of day | | |
| 00:00:07:59 | 212 (27.1) | 202 (27.3) |
| 08:00:15:59 | 252 (32.3) | 196 (26.5) |
| 16:00:23:59 | 317 (40.5) | 341 (46.1) |
| Instrument used | | |
| Sharp object | 474 (60.7) | 89 (12.0) |
| Blunt object | 112 (14.3) | 70 (9.5) |
| Firearm | 29 (3.7) | 89 (12.0) |
| Push/kick/punch | 115 (14.7) | 309 (41.8) |
| Human bite | 6 (0.8) | 2 (0.3) |
| Choking/strangulation | 2 (0.3) | 8 (1.1) |
| Fire burn | 2 (0.3) | 5 (0.7) |
| Other burn | 3 (0.4) | 2 (0.3) |
| Poisoning | 0 (0.0) | 56 (7.6) |
| Unknown | 38 (4.9) | 109 (14.7) |
| Type of violence | | |
| Rape/sexual | 16 (2.0) | 67 (9.1) |
| Property crime related | 290 (37.0) | 250 (33.8) |
| Interpersonal | 266 (44.8) | 422 (57.1) |
| Gang related | 84 (10.7) | |
| Unknown | 125 (16.0) | † |
| Area | | |

Continued

**Table 1** Continued

| Variable | Health data (N=781) n (%) | Police data (N=739) n (%) |
|---|---|---|
| Enkanini | 41 (5.2) | 9 (1.2) |
| Harare | 85 (10.9) | 66 (8.9) |
| Ilitha Park | 21 (2.7) | 42 (5.7) |
| Khaya | 30 (3.8) | 74 (10.0) |
| Kuyasa | 39 (5.0) | 45 (6.1) |
| Makaya | 43 (5.5) | 37 (5.0) |
| Makhaza | 85 (11.0) | 96 (13.0) |
| Site B | 283 (36.2) | 243 (32.9) |
| Site C | 118 (15.1) | 91 (12.3) |
| Town 2 | 36 (4.6) | 36 (4.9) |
| Triage scores‡ | | |
| Red | 65 (8.3) | – |
| Orange | 251 (32.1) | – |
| Yellow | 380 (48.6) | – |
| Green | 68 (8.7) | – |
| Blue | 1 (0.12) | – |
| Unknown | 18 (2.2) | – |
| Alcohol use‡ | | |
| Yes | 444 (56.8) | – |
| No | 284 (36.4) | – |
| Unknown | 53 (6.8) | – |

*Three hundred and fifty-eight police records had missing values on gender and age.
†For type of violence, the police data did not record an 'unknown or gang-related' category.
‡Triage score and alcohol use data were only available for the health dataset.

the triage code). There were minimal differences in location and timing of the violent episode between reporting and non-reporting patients. The one significant difference in timing of the violence episode was the almost double number of episodes reported on Sunday among non-reporting patients when compared with reporting patients (36.1% vs 20.2%, p=0.017).

## DISCUSSION

One of the key findings of a 2014 systematic review of the feasibility and effectiveness of community-level interventions involving ED data sharing to reduce alcohol-related violence was that data-sharing protocols can be cheaply and easily implemented into modern ED triage systems.[18] There was general consensus among studies that this could be done with minimal cost, staff workload burden and impact to patient safety. Additionally, there was minimum risk to patient service and anonymity, risk of harm displacement to other licensed venues or

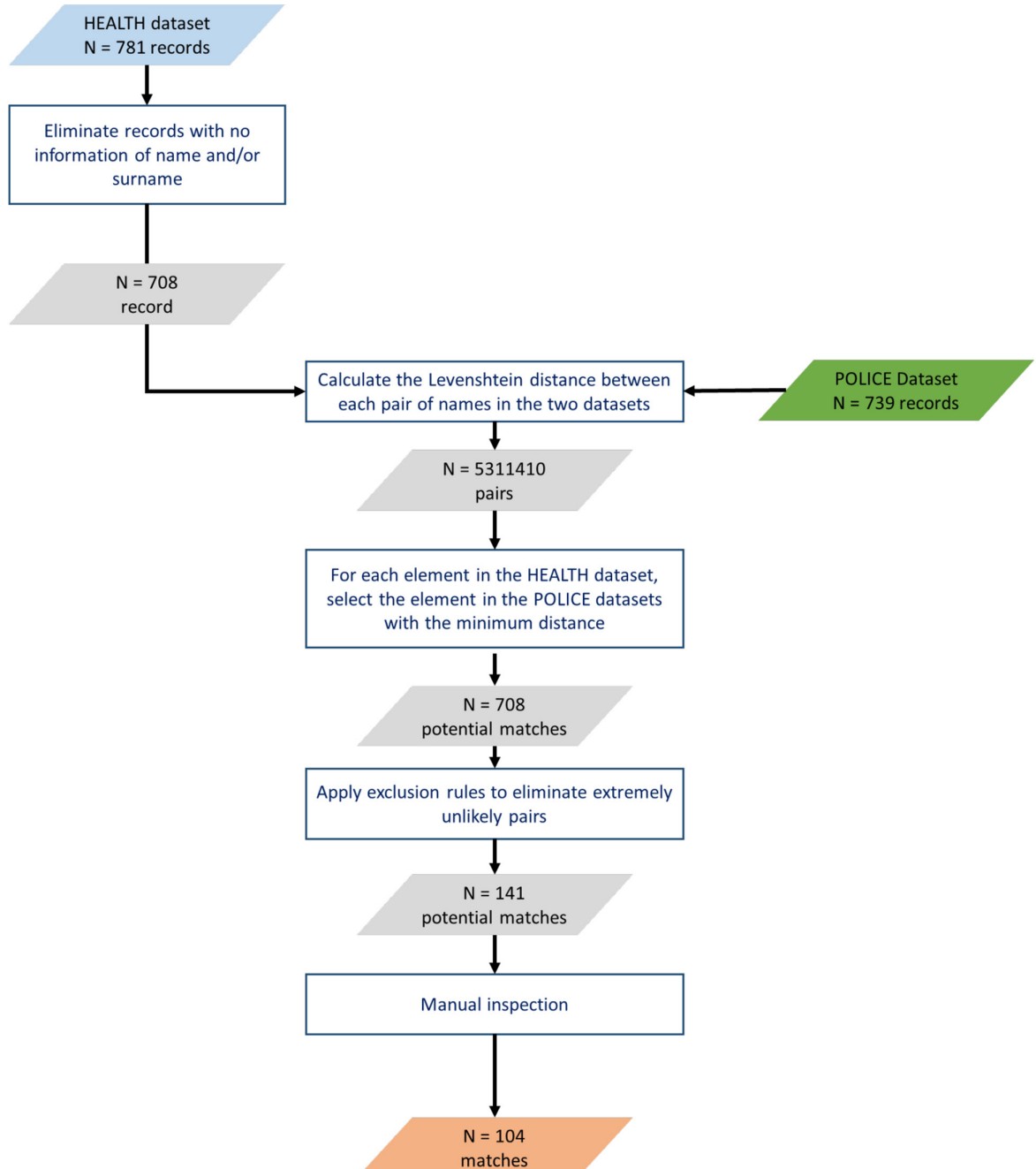

**Figure 3** Personal identifier matching between datasets.

increase to length of patient stay.[18] The Cardiff Model is cost-effective with one study finding that for every US$1 spent, nearly US$15 in health system and over US$19 in criminal justice system costs are saved.[19]

To the best of our knowledge, this is the first study to attempt direct name matching between the health and the police to quantify the level of under-reporting of interpersonal violence-related incidents. The matching method using personal identifiers suggest that only a small fraction (approximately 15%) of patients being treated for interpersonal violence at local health facilities will go on to report these as crimes to local police stations within a week. Conversely, only a small proportion of those who

report a violent crime to the police go on to present to a health facility with an injury within a week.

The proportion of cases in this study of injuries arising from interpersonal violence which are not reported to the police (approximately 82%) is consistent with the findings of the Cardiff model which found that one-half to two-thirds of violence which results in hospital treatment is not known to the police.[2] This finding is much lower when compared with other regions including in developed countries. According to a 2016 US Department of Justice report, many crimes go unreported to law enforcement, including 53% of violent crime in 2015; 58% of simple assaults in 2015 and 43% of violent crime involving an injury in 2015.[20]

**Table 2** Adjusted crime profile comparing police versus police and health combined data

| Variable | Police data (N=739) | | Combined data (N=1343) | |
| --- | --- | --- | --- | --- |
| | Rank | n (%) | Rank | n (%) |
| Gender* | | | | |
| Males | 1 | 196 (51.4) | 1 | 640 (65.0) |
| Females | 2 | 185 (48.6) | 2 | 345 (35.0) |
| Age (years)* | | | | |
| 16–24 | 2 | 85 (22.3) | 2 | 274 (27.8) |
| 25–34 | 1 | 130 (34.1) | 1 | 394 (40.0) |
| 35–44 | 3 | 72 (18.9) | 3 | 173 (17.6) |
| 45–54 | 4 | 56 (14.7) | 4 | 87 (8.8) |
| 55+ | 5 | 38 (10.0) | 5 | 57 (5.8) |
| Day of week | | | | |
| Monday | 3 | 103 (13.9) | 3 | 178 (13.2) |
| Tuesday | 6 | 72 (9.7) | 6 | 107 (8.0) |
| Wednesday | 5 | 75 (10.1) | 5 | 109 (8.1) |
| Thursday | 7 | 64 (8.7) | 7 | 96 (7.1) |
| Friday | 4 | 100 (13.5) | 4 | 155 (11.5) |
| Saturday | 1 | 165 (22.3) | 2 | 320 (23.8) |
| Sunday | 2 | 160 (21.7) | 1 | 378 (28.1) |
| Time of day | | | | |
| 00:00:07:59 | 2 | 202 (27.3) | 3 | 362 (26.9) |
| 08:00:15:59 | 3 | 196 (26.5) | 2 | 390 (29) |
| 16:00:23:59 | 1 | 341 (46.1) | 1 | 591 (43.9) |
| Instrument used | | | | |
| Sharp object | 3 | 89 (12.0) | 1 | 476 (35.4) |
| Blunt object | 4 | 70 (9.5) | 5 | 155 (11.5) |
| Firearm | 3 | 89 (12.0) | 4 | 111 (8.3) |
| Push/kick/punch | 1 | 309 (41.8) | 2 | 383 (28.5) |
| Human bite | 8 | 2 (0.3) | 9 | 5 (0.4) |
| Choking/strangulation | 6 | 8 (1.1) | 7 | 9 (0.7) |
| Fire burn | 7 | 5 (0.7) | 8 | 7 (0.5) |
| Other burn | 8 | 2 (0.3) | 10 | 5 (0.4) |
| Poisoning | 5 | 56 (7.6) | 6 | 56 (4.2) |
| Unknown | 2 | 109 (14.7) | 3 | 136 (10.1) |
| Type of violence | | | | |
| Rape/sexual | 3 | 67 (9.1) | 3 | 74 (5.5) |
| Property crime related | 2 | 250 (33.8) | 2 | 489 (36.4) |
| Interpersonal† | 1 | 422 (57.1) | 1 | 780 (58.1) |
| Area | | | | |
| Enkanini | 10 | 9 (1.2) | 8 | 41 (3.1) |
| Harare | 5 | 66 (8.9) | 4 | 129 (9.6) |
| Ilitha Park | 7 | 42 (5.7) | 7 | 59 (4.4) |
| Khaya | 4 | 74 (10.0) | 4 | 96 (7.1) |
| Kuyasa | 6 | 45 (6.1) | 5 | 74 (5.5) |
| Makaya | 8 | 37 (5.0) | 6 | 70 (5.2) |
| Makhaza | 2 | 96 (13.0) | 3 | 163 (12.1) |

**Table 2** Continued

| Variable | Police data (N=739) | | Combined data (N=1343) | |
| | Rank | n (%) | Rank | n (%) |
|---|---|---|---|---|
| Site B | 1 | 243 (32.9) | 1 | 461 (34.3) |
| Site C | 3 | 91 (12.3) | 2 | 188 (14.0) |
| Town 2 | 9 | 36 (4.9) | 7 | 62 (4.6) |

*Three hundred and fifty-eight police records had missing values on gender and age.
†For purposes of comparison, 'gang-related' cases from the health data were included as 'interpersonal violence' cases. Additionally, the 'unknown' category was included as 'interpersonal violence' for purposes of comparison and ranking.

Possible reasons for the high level of under-reporting (30%+ discrepancy between the level of under-reporting in this study (82%) and the Cardiff model (50%)) can be found in the report of the 2012 Khayelitsha Commission of inquiry (also known as the O'Regan/Pikoli Commission) which investigated allegations of police inefficiency in Khayelitsha and the breakdown in relations between the Khayelitsha community and the police. Poor policing conditions characterised by police in the area being generally lower-ranked, poorly equipped, few in number, underqualified and reluctant to open cases led to breakdown in trust with the community resulting in the lack of reporting.[21] Other possible reasons for under-reporting include the fear of reprisal from perpetrators, being turned away by police when presenting at the police station to report a crime, stigma/shame or not perceiving an injury arising from interpersonal violence as a crime.

### Type of injury
The majority of police-reported crimes are related to assault with body part (push/kick/punch). Victims of knife crimes represented in the health data, which often cause worse injuries requiring medical intervention, are not reporting these incidents to police. Gang/property crime-related episodes were more common among non-reporting patients, which may be due to the lack of confidence in the police's ability to protect them from reprisal, or if the victim perceived the gangs as being protected by or working with the police, in a known context of mistrust of police in the community of Khayelitsha.[21] Additionally, the under-reporting of knife wielding gang-related violence incidents in police statistics may be a contributing factor. This is supported by the lower proportion of younger male victims (age 16–24 years) reporting crime compared with those receiving treatment for an injury arising from interpersonal violence (31.2% vs 39.4%, p=0.017).

The scale in difference of sharp object injuries presenting at police (89 cases) and health facilities (474) means that 385 victims of violence treated for a sharp object injury at a health facility did not go on to report the crime to the police, which if added to the official police statistics would represent a 52% increase in all violent crimes reported. This provides evidence for an excess burden of injuries arising from interpersonal violence which is not being reported to police—a situation that is consistent with the premise of the Cardiff model.

### LIMITATIONS
One of the limitations of this study is that approximately 10% of health records lacked names and were thus excluded from the matching process. A 2013 South African study in missing medical records found that up to 8% of hospital folders had the wrong names and that 17% of hospital folders had names without folder numbers.[22]

For each 1-week data collection period, we should have extended the police data collection period to account for possible late reporting. Our results may, therefore, underestimate potential matching. In attempting to quantify the extent of this under-reporting, we considered the difference between time of presentation to the clinical facility and time recorded in the SAPS database and the percentage-matched cases by day of week (online supplemental table 1).

In most cases, there was a modest difference between time of reporting (mean 7.9 hours, median 2.6 hours, 90th percentile 23 hours), and percent matching on the final day of the reporting period (22.7% on Wednesday) was higher than the average of 17.5%. We concluded that the extent of under-reporting was negligible and would not materially affect our results.

No cases of poisoning were recorded in the health data. The current referral practice is for Michael M and Site B to refer their poisoning cases to KDH. At KDH, most poisoning cases are triaged to the Medical department and thus these cases would not be recorded in the health data. The exception being cases of poisoning due to a corrosive substance such as bleach, which would be triaged to the Trauma/Surgical department due to the possible need for a surgical therapeutic intervention. It is a recommendation of study authors that the poisoning data from medical emergencies be included to inform this work going forward.

One of the weaknesses of this study is that patients who presented to a health facility but only lodged criminal cases later outside of the 1-week study period over the 5 years would not be included in the police dataset and therefore not be included in the matching process.

**Table 3** Difference in demographic and clinical characteristics and in type of offence between patients who did/did not report the violence to police

| Variable | Patients who did not report the violence | Patients who reported the violence | All patients* | P value† |
|---|---|---|---|---|
| | N=604 | N=104 | N=708 | |
| | n (%) | n (%) | n (%) | |
| Gender | | | | |
| Male | 444 (73.5) | 44 (42.3) | 488 (68.9) | |
| Female | 160 (26.5) | 60 (57.7) | 220 (31.1) | <0.001 |
| Age (years) | | | | |
| 16–24 | 188 (31.2) | 41 (39.4) | 229 (32.4) | 0.017 |
| 25–34 | 264 (43.8) | 34 (32.7) | 298 (42.1) | |
| 35–44 | 101 (16.7) | 16 (15.4) | 117 (16.5) | |
| 45–54 | 31 (5.1) | 12 (11.5) | 43 (6.1) | |
| 55+ | 20 (3.2) | 1 (1.0) | 21 (2.8) | |
| Day of week | | | | |
| Monday | 75 (12.4) | 14 (13.5) | 89 (12.6) | 0.017 |
| Tuesday | 35 (5.8) | 12 (11.5) | 47 (6.6) | |
| Wednesday | 34 (5.6) | 10 (9.6) | 44 (6.2) | |
| Thursday | 32 (5.3) | 5 (4.8) | 37 (5.2) | |
| Friday | 55 (9.1) | 15 (14.4) | 70 (9.9) | |
| Saturday | 155 (25.7) | 27 (26) | 182 (25.7) | |
| Sunday | 218 (36.1) | 21 (20.2) | 239 (33.8) | |
| Time of day | | | | |
| 00:00:07:59 | 160 (26.4) | 30 (28.9) | 190 (26.8) | 0.267 |
| 08:00:15:59 | 194 (32.2) | 39 (37.4) | 233 (33) | |
| 16:00:23:59 | 250 (41.4) | 35 (33.6) | 285 (40.3) | |
| Instrument | | | | |
| Sharp object | 387 (64.1) | 41 (39.4) | 428 (60.5) | <0.001 |
| Blunt object | 85 (14.1) | 18 (17.3) | 103 (14.5) | |
| Firearm | 22 (3.6) | 5 (4.8) | 27 (3.8) | |
| Push/kick/punch | 74 (12.3) | 31 (29.8) | 105 (14.8) | |
| Human bite | 3 (0.5) | 1 (1) | 4 (0.6) | |
| Choking/strangulation | 1 (0.2) | 1 (1) | 2 (0.3) | |
| Burns | 5 (0.8) | 0 (0) | 5 (0.7) | |
| Unknown | 27 (4.5) | 7 (6.7) | 34 (4.8) | |
| Type of violence | | | | |
| Rape/sexual | 7 (1.2) | 9 (8.7) | 16 (2.3) | |
| Property crime related‡ | 239 (39.6) | 26 (25.0) | 265 (37.4) | |
| Interpersonal | 199 (33.0) | 45 (43.2) | 244 (34.5) | <0.001 |
| Gang related | 69 (11.4) | 6 (5.8) | 75 (10.6) | |
| Unknown | 90 (14.9) | 18 (17.3) | 108 (15.3) | |
| Area | | | | |
| Enkanini | 32 (5.3) | 4 (3.8) | 36 (5.1) | |
| Harare | 63 (10.4) | 14 (13.5) | 77 (10.9) | |
| Ilitha Park | 17 (2.8) | 1 (1) | 18 (2.5) | |
| Khaya | 22 (3.6) | 4 (3.8) | 26 (3.7) | |

Continued

**Table 3** Continued

| Variable | Patients who did not report the violence N=604 n (%) | Patients who reported the violence N=104 n (%) | All patients* N=708 n (%) | P value† |
|---|---|---|---|---|
| Kuyasa | 29 (4.8) | 6 (5.8) | 35 (4.9) | 0.689 |
| Makaya | 33 (5.5) | 7 (6.7) | 40 (5.6) | |
| Makhaza | 67 (11.1) | 13 (12.5) | 80 (11.3) | |
| Site B | 218 (36.1) | 38 (36.5) | 256 (36.2) | |
| Site C | 97 (16.1) | 10 (9.6) | 107 (15.1) | |
| Town 2 | 26 (4.3) | 7 (6.7) | 33 (4.7) | |
| Triage code | | | | |
| Red | 47 (7.8) | 7 (6.7) | 54 (7.6) | |
| Orange | 204 (33.8) | 23 (22.1) | 227 (32.1) | 0.278 |
| Yellow | 290 (48) | 58 (55.8) | 348 (49.2) | |
| Green | 53 (8.8) | 8 (7.7) | 61 (8.6) | |
| Blue | 1 (0.2) | 0 (0) | 1 (0.1) | |
| Unknown | 9 (1.5) | 8 (7.7) | 17 (2.4) | |
| Alcohol use | | | | |
| Yes/suspected | 356 (58.9) | 49 (47.1) | 405 (57.2) | |
| No | 205 (33.9) | 50 (48.1) | 255 (36) | 0.012 |
| Unknown | 43 (7.1) | 5 (4.8) | 48 (6.8) | |

*For which the name and surname could be retrieved from the Clinicom database.
†Statistical test for the existence of differences between reporting and non-reporting patients.
‡Property crime related refers to the type of violence that could not be categorised under rape/sexual, interpersonal violence, gang violence or unknown. Cases in this category included violence initiated to facilitate property theft from an individual.

Similarly, if victims of injuries arising from interpersonal violence presented to the police first and did not present to the health facilities for treatment within the 1-week study period, they were also not included in the matching process. The inclusion of these late reporting cases could decrease the overall proportion of interpersonal violence-related injuries that were reported to the police. Finally, patients experiencing interpersonal violence and who sought private healthcare or presented directly to the hospitals outside of the district (such as the tertiary referral hospital, Tygerberg Hospital) would not be included in the analyses. This may increase or decrease the estimate of concordance depending on whether these patients reported to the police.

## IMPLICATIONS FOR POLICY AND INTERVENTIONS

Murder and violence are often linked to alcohol with a 2016 analysis of murder dockets finding that in up to 48% of murders, the victim or perpetrator was intoxicated.[23] Alcohol use was higher in patients who did not report violence than patients who did report violence to the police. This is of public health importance as alcohol has been identified as a major risk factor for and predictor of violence and is amenable to policy change to reduce alcohol harms. The current evidence base for measuring alcohol use within crime data will underestimate the extent of the problem because of the under-reporting in routine crime statistics highlighted by this research.

Furthermore, with regard to local municipal alcohol policy on liquor outlet licences,[24] reasons for limitation of liquor outlet licences may include 'safety and social problems and/or concerns arising from alcohol abuse, alcohol-related crimes, illegal criminal activity, etc, using data derived from the South African Police Service'. The local alcohol policy is heavily reliant on the SAPS data to guide license approvals, which this research has shown to be incomplete and flawed in its underestimation of interpersonal violence at the community level. Alcohol policy decisions made by government at any level (local, district, provincial and national) that is informed by this data may aggravate alcohol-related harms.

The level of under-reporting exposed by the merging of health and police datasets and matching exercise may speak to a larger question of police resourcing within Khayelitsha. The Khayelitsha commission of inquiry highlighted the insufficient police human resources within police stations in Khayelitsha as one of the reasons for community mistrust of the police.[21] The current SAPS resource allocation model is based on the Theoretical Human Resource Requirement and fixed establishment

approach which was developed in 2012.[25] The tool takes into consideration annual reported crime per police station,[25] which suggests that current police human resources deployed in Khayelitsha may be severely under-resourced to deal with the 'real' burden of violence, when unreported health data are considered.

The validity of self-reported drinking using ED data from 16 countries reported findings suggesting that the overall validity of self-reported alcohol consumption before injury is high (88%) compared with a blood alcohol concentration estimate obtained at the time of admission to the ED, with a caveat that demographic, drinking and injury characteristics may affect validity.[26]

This study provides evidence showing that interpersonal violence at the community level cannot be estimated accurately relying solely on reported crimes to the police services. Injuries arising from interpersonal violence treated at health facilities must be included to more accurately define the level of interpersonal violence within a community.

Implementation of the Hospital Emergency Centre Tracking Information System, an electronic data collection system for EDs, is currently underway at the Michael Mapongwana CHC and KDH. This will improve the monitoring of injuries arising from interpersonal violence (the majority of which are not reported to the police) and provide a method for the evaluation of violence prevention interventions. Additionally, this will provide the data infrastructure required for the potential implementation of the Cardiff model locally.

Preliminary findings from this research were shared with the Whole of Society Approach (WOSA) intergovernmental task team that operates in the community of Khayelitsha, with a mandate to reduce interpersonal violence. The full findings will be shared with the WOSA stakeholders as preliminary work to inform the full implementation of the Cardiff model in Khayelitsha early next year.

## CONCLUSION

The matching exercise described above represents the first attempt within South Africa to match patients at health facilities treated for injuries arising from interpersonal violence with those of victims reporting violent crimes to police stations within the same community and time period. This matching study suggests that the majority of injuries arising from interpersonal violence presenting at health facilities in Khayelitsha are not reported to the police in Khayelitsha.

This study has broader implications regionally and nationally for the surveillance of injuries arising from interpersonal violence, for the police definition and surveillance of community interpersonal violence, for community policing intelligence development (improving the configuration of violence heat maps on a real-time basis) and finally for police resource utilisation and distribution, which should, in turn, impact positively on reducing crime and violence in the community, and reduce the burden on the Health services. The implementation of the Cardiff data-sharing model should be adopted to inform violence reduction locally.

**Contributors** AJ and RM developed the research proposal and were involved in the analysis and interpretation of the data. AC was involved with the analysis of the data including the matching protocol. LL and TO revised the work for content and approved the final version to be published. Aj is the study guarantor.

**Funding** This work was supported by the International Development Research Centre (IDRC), Canada, and the UK Government's Department for International Development (DFID) (grant no 107329-001) and the Western Cape Government Department of Health (grant no WCDOH428/2013). The degree from which this study emanated was funded by the South African Medical Research Council (SAMRC) under the SAMRC Clinician Researcher (MD PhD) Scholarship Programme. RM is funded by the SAMRC. TO is supported by the National Institute for Health Research (NIHR) (16/137/34) using UK aid from the UK Government to support global health research.

**Disclaimer** The views expressed in this publication are those of the authors and not necessarily those of the National Institute for Health and Care Research or the UK Department of Health and Social Care or SAMRC.

**Map disclaimer** The inclusion of any map (including the depiction of any boundaries therein), or of any geographic or locational reference, does not imply the expression of any opinion whatsoever on the part of BMJ concerning the legal status of any country, territory, jurisdiction or area or of its authorities. Any such expression remains solely that of the relevant source and is not endorsed by BMJ. Maps are provided without any warranty of any kind, either express or implied.

**Competing interests** None declared.

**Patient and public involvement** Patients and/or the public were not involved in the design, conduct, reporting or dissemination plans of this research.

**Patient consent for publication** Not applicable.

**Ethics approval** This study involves human participants. Ethical permission to conduct this research was granted by the University of Cape Town Human Research Ethics Committee (UCT HREC 861/2016) and the South African Police Services Research Division (Application no. 3/34/2). With regard to the HST data, informed consent was obtained only from acute injury cases and where given, these individuals were interviewed by data collectors, with only acute injury cases having the full data collection tool completed. Informed consent was granted by Groote Schuur Hospital for permission to access the Clinicom patient database. No minors were interviewed as part of this study. Participants gave informed consent to participate in the study before taking part.

**Provenance and peer review** Not commissioned; externally peer reviewed.

**Data availability statement** No data are available. No data are available

**ORCID iDs**
Ardil Jabar http://orcid.org/0000-0003-1040-3869
Leslie London http://orcid.org/0000-0003-1297-2758
Annibale Cois http://orcid.org/0000-0002-7014-6510

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
