## [Reviewer comments · BMJ Open]

ARTICLE DETAILS

TITLE (PROVISIONAL)	MATCHING STUDY USING HEALTH AND POLICE DATASETS FOR CHARACTERISING INTERPERSONAL VIOLENCE IN THE COMMUNITY OF KHAYELITSHA, SOUTH AFRICA 2013-2015
AUTHORS	Jabar, Ardil; Oni, Tolu; London, Leslie; Cois, Annibale; Matzopoulos, Richard

VERSION 1 – REVIEW

REVIEWER	Elinore Kaufman University of Pennsylvania, Surgery
REVIEW RETURNED	04-Feb-2021

GENERAL COMMENTS	This paper assesses the potential of Cardiff-style data sharing on violence in Khayelitsha. This is a well done, well presented study. I have a few questions that I hope can strengthen the manuscript. 1. What are your thoughts on patient privacy? This is the first name-based match because of restrictions on disclosure that are in place to protect patients.2. Relatedly, how do you approach the risks that patients may accrue when their data is shared with police? I suspect that even anonymous data sharing could lead to identification in some cases.3. Did you consider asking patients directly if they had reported events to the police?4. The Cardiff model is only effective if the information is used to prevent violence or crime. How do you plan to implement these findings? Are there opportunities besides increased intensity of policing that might be useful and cause less harm?5. Table 3: What does “crime-related” mean? IF you expect them to report to the police, shouldn't it all be crime-related?6. Table 3: Consider making this more readable by excluding some of the variables that might be relevant for matching but not for interpretation—such as the many divisions of time of day and the very rare mechanisms of injury.7. Figure 2: the use of “condition 1” etc is confusing. Please state the reasons for exclusion in the figure or legend so that the figure can stand alone.
--

REVIEWER	Rachel Odes University of California San Francisco, Institute for Health Policy Studies
REVIEW RETURNED	26-Apr-2021

GENERAL COMMENTS	This is a thoughtful, compelling manuscript drawing on health and police datasets to describe rates of crime and injuries due to violence in Khayelitsha, Cape Town, South Africa. There are multiple strengths of this submission, including use of an appropriate
---

	theoretical construct and a well-thought-out methodological approach. In addition, the research takes on a meaningful problem and offers some concrete implications for policy. I believe the manuscript could be strengthened with several revisions described below: Abstract: In the methods section of the abstract, the authors write: “781 cases of injuries arising from interpersonal violence seen at health facilities were matched with 739 violence-related crimes...” Can the word compared be used instead of matched? Or state that matching was attempted? It is somewhat confusing because they go on to state in results that 104 cases were actually matched. Methods: Data sources: The most significant concern is the rationale for using the same one-week periods of time for both the police and health datasets. While it seems self-evident that victims of violent crime would present for medical care immediately, it would be helpful to cite some literature showing that one week is a reasonable period to assume that crimes would be reported to the police. Case selection: Elaboration on the exclusion of cases where the victim was under 16 y.o. would be helpful here. The authors state that they have set this cutoff due to the age of consent for sexual behavior, but it is not entirely clear how this relates to the research interest. It would be helpful to have a little more information. Results: Tables 1 and 2: There is a lot of detail presented in these tables. While the categories are useful for some of these variables, the comparison of specific times of occurrence/report seem less meaningful. Could some of these findings be collapsed into a range? (Eg 6- or 8-hour ranges instead of 2-hour ranges?) Similarly, with days of the week: could ranges be substituted? Figure 2: This figure communicates essential information about how cases were selected for inclusion. Is it possible to provide a very abbreviated explanation for each condition included, or to add a key for the entire figure? Discussion: If the same, one-week periods are used for both datasets, might it be possible that victims of violence presenting to the hospital on Saturday or Sunday might not report the crime to police until the following Monday? If this is the case, then the health records from the end of the week would be much less likely to be matched than those if the beginning of the week. Table 3 seems to indicate that Sat/Sun have a far larger number of non-reported crimes when compared to other days of the week and there is no discussion of this finding and the potential impact the study design may be having on the results. Thank you for the opportunity to review this thought-provoking work.
--	--

VERSION 1 – AUTHOR RESPONSE

Reviewer 1

1. What are your thoughts on patient privacy? This is the first name-based match because of restrictions on disclosure that are in place to protect patients.

As a health care professional, patient privacy is respected and upheld in accordance with medical ethics guidelines and national legislation such as the South African Protection of Personal Information (POPI) Act. In keeping with these principles, study researchers were blinded to personal identifier data. The study Biostatistician received both health and police datasets, conducted the identifier matching process, and deleted all data permanently following this process.

We have clarified this in the manuscript with the following text in the ethics section:

“The study Biostatistician received both health and police datasets, conducted the identifier matching process, and deleted all data permanently following this process.”

2. Relatedly, how do you approach the risks that patients may accrue when their data is shared with police? I suspect that even anonymous data sharing could lead to identification in some cases.

As per the guidelines of Cardiff model, only anonymised data is used and data is aggregated and shared in a one-month delay to the relative stakeholders.

We have clarified this in the manuscript with the following text in the ethics section:

“By aggregating the data, removing personal identifiers, geolocating cases to residential suburb and not street address, and employing a one-month delay of data dissemination, every technical effort is made to prevent the identification of individual cases and protect individual identity.”

3. Did you consider asking patients directly if they had reported events to the police?

While we acknowledge that asking patients if they had reported events to the police would have been a useful and helpful variable to collect, this study used secondary data from a previous research study conducted by the Health Systems Trust between 2013 to 2015. As such, no primary data collection was conducted. We have included this limitation in the manuscript, in the data sources section, with text that reads as follows:

“The health data used in this study were originally collected by a non-profit research organisation, the Health Systems Trust (HST), for a study commissioned by the Western Cape Department of Health (DOH). Data were collected for five 1-week-long cross-section periods between 2013 to 2015.”

4. The Cardiff model is only effective if the information is used to prevent violence or crime. How do you plan to implement these findings? Are there opportunities besides increased intensity of policing that might be useful and cause less harm?

We have included new text in the manuscript, in the ‘implications for policy and interventions’ section, as follows:

“Preliminary findings from this research were shared with the Whole of Society Approach (WOSA) intergovernmental task team that operates in the community of Khayelitsha, with a

mandate to reduce interpersonal violence. The full findings will be shared with the WOSA stakeholders as preliminary work to inform the full implementation of the Cardiff model in Khayelitsha later this year.”

Furthermore, the following two implications are described further in the discussion section:

Other opportunities with regards to policing include the appropriate human resourcing of policing personnel and the distribution of liquor licenses within the community. As both of these are calculated based on number of crimes reported within the community, this research has shown that this is based on invalid data and thus both police human resources and liquor license distribution are targeted areas for reformation.

Opportunities for the health sector include the advocacy and implementation of electronic data collection tools at Emergency Departments within clinics and day hospitals in Khayelitsha. The importance of the health sector is highlighted in its role in contributing to the more accurate description of the burden of interpersonal violence prevention at the community level.

5. Table 3: What does “crime-related” mean? IF you expect them to report to the police, shouldn't it all be crime-related?

With regard to Table 3, crime-related cases in this category included violence initiated to facilitate property theft from an individual, i.e. the type of violence that could not be categorised under rape/sexual, interpersonal violence, gang violence or unknown. This is set out in some detail in the variables section of the methods as follows:

“all types of robbery, which involve an interpersonal interaction with an implicit threat (i.e. common robbery; house robbery; robbery with a weapon or instrument other than a firearm; robbery and attempted robbery with a fire-arm; robbery and attempted robbery at business premises; carjacking [stealing an occupied motor-vehicle] (crime-related)”

For the sake of clarity, and to avoid the possible misconception of other categories of violence not been considered criminal, we have renamed this category “property crime-related” throughout the document.

6. Table 3: Consider making this more readable by excluding some of the variables that might be relevant for matching but not for interpretation—such as the many divisions of time of day and the very rare mechanisms of injury.

The time variable has been revised to 8-hour ranges. The poisoning variable has been removed. All the burns cases have been combined into a single category.

7. Figure 2: the use of “condition 1” etc is confusing. Please state the reasons for exclusion in the figure or legend so that the figure can stand alone.

Figure 2 has been revised with conditions described in the figure key.

Reviewer: 2

1. Abstract:

In the methods section of the abstract, the authors write: “781 cases of injuries arising from interpersonal violence seen at health facilities were matched with 739 violence-related crimes...” Can the word compared be used instead of matched? Or state that matching was attempted? It is somewhat confusing because they go on to state in results that 104 cases were actually matched.

The word “matched” has been replaced by “compared” within the methods section in abstract.

2. Methods:

Data sources: The most significant concern is the rationale for using the same one-week periods of time for both the police and health datasets. While it seems self-evident that victims of violent crime would present for medical care immediately, it would be helpful to cite some literature showing that one week is a reasonable period to assume that crimes would be reported to the police.

We agree that this is a clear limitation of this study. As the health data was secondary data used from the Health Systems Trust (HST) original study, as indicated in the manuscript in the data sources section, we were limited to the one-week data collection periods employed in the original study. The SAPS data collection period should have been extended to account for this.

We tried to assess the potential impact by analysing the matching proportion according to the day of the week (Table 1). All data collection weeks started on a Thursday at 07:00. The average difference between time of presentation to the clinical facility and time recorded in the SAPS database (absolute value) was 7.9 hours.

Table 1. Matching proportion per day of week of presentation to the clinical facility

Day	Matching proportion [%]
Thursday	11.1
Friday	19.5
Saturday	13.2
Sunday	8.0
Monday	14.4
Tuesday	24.0
Wednesday	20.8

The data (Table 1) does not seem to show a declining rate, and most cases are reported within 24 hours. We conclude that if certainly, the problem exists, there is no indication of a large bias in our results.

3. Case selection: Elaboration on the exclusion of cases where the victim was under 16 y.o. would be helpful here. The authors state that they have set this cutoff due to the age of consent for sexual behavior, but it is not entirely clear how this relates to the research interest. It would be helpful to have a little more information.

This study employed secondary data from a previous research study conducted by the Health Systems Trust between 2013 to 2015. The exclusion of cases where the victim was under 16 years old were made by the researchers in the original primary study. Victims under 16 years old are considered children under South African law and the focus of the primary study was adults. The sentence describing the reason for age cut-off, has been removed from the case selection section.

4. Results:

Tables 1 and 2: There is a lot of detail presented in these tables. While the categories are useful for some of these variables, the comparison of specific times of occurrence/report seem less meaningful. Could some of these findings be collapsed into a range? (Eg 6- or 8-hour ranges instead of 2-hour ranges?) Similarly, with days of the week: could ranges be substituted?

Table 1 has been revised and times have been collapsed into 8-hour ranges as suggested. Days of week have not been revised as this would remove the individual day burden of injuries in Table 1 which may hide the weekend patten of burden of injury, and remove the ranking system for days in Table 2.

5. Figure 2: This figure communicates essential information about how cases were selected for inclusion. Is it possible to provide a very abbreviated explanation for each condition included, or to add a key for the entire figure?

Figure 2 has been revised accordingly with the addition of a key for the figure.

6. Discussion:

If the same, one-week periods are used for both datasets, might it be possible that victims of violence presenting to the hospital on Saturday or Sunday might not report the crime to police until the following Monday?

We agree that this is a clear limitation of this study. As the health data was secondary data used from the HST original study, we were limited to the one-week data collection periods employed in the original study. The SAPS data collection period should have been extended to account for this.

7. If this is the case, then the health records from the end of the week would be much less likely to be matched than those if the beginning of the week. Table 3 seems to indicate that Sat/Sun have a far larger number of non-reported crimes when compared to other days of the week and there is no discussion of this finding and the potential impact the study design may be having on the results.

This has been addressed under Reviewer 2, comment 2 above.

VERSION 2 – REVIEW

REVIEWER	Elinore Kaufman University of Pennsylvania, Surgery
REVIEW RETURNED	28-Jan-2022

GENERAL COMMENTS	The revisions are appropriate.
--------------------------------

REVIEWER	Rachel Odes University of California San Francisco, Institute for Health Policy Studies
REVIEW RETURNED	11-Feb-2022

GENERAL COMMENTS	2/11/22: Thank you for the opportunity to review the revision of this manuscript. The author's response has addressed most of concerns identified in the initial review. One concern I raised, in the methods section / data sources, was addressed by the author's response but did not lead to any changes in the manuscript as far as I can tell. I would encourage the author to include some additional information about the typical time for reporting crimes (they assert in their response that most are reported within 24 hours).
--

VERSION 2 – AUTHOR RESPONSE

Reviewer 2

1. Thank you for the opportunity to review the revision of this manuscript. The author's response has addressed most of concerns identified in the initial review. One concern I raised, in the methods section / data sources, was addressed by the author's response but did not lead to any changes in the manuscript as far as I can tell. I would encourage the author to include some additional information about the typical time for reporting crimes (they assert in their response that most are reported within 24 hours).

1. Discussion:

The following paragraph has been added to the Discussion section under limitations, with a Supplemental Table 1 provided as an appendix.

For each one-week data collection period we should have extended the police data collection period to account for possible late reporting. Our results may therefore underestimate potential matching. In attempting to quantify the extent of this underreporting we considered the difference between time of presentation to the clinical facility and time recorded in the SAPS database and the percentage matched cases by day of week (Supplemental Table 1).

In most cases there was a modest difference between time of reporting (mean 7.9 hours, median 2.6 hours, 90th percentile 23 hours) and percent matching on the final day of the reporting period (22.7% on Wednesday) was higher than the average of 17.5%. we concluded that the extent of underreporting was negligible and would not materially affect our results (Supplemental Table 1).

Supplemental Table 1. Matching proportion per day of week of presentation to the clinical facility

DAY MATCHED TOTAL PERCENT (%)

Monday 14 89 15,7

Tuesday 12 47 25,5

Wednesday 10 44 22,7

Thursday 5 37 13,5

Friday 15 70 21,4

Saturday 27 182 14,8

Sunday 21 239 8,8